# Modulating the Heat Stress Response to Improve Hyperthermia-Based Anticancer Treatments

**DOI:** 10.3390/cancers13061243

**Published:** 2021-03-12

**Authors:** Enzo M. Scutigliani, Yongxin Liang, Hans Crezee, Roland Kanaar, Przemek M. Krawczyk

**Affiliations:** 1Department of Medical Biology, Amsterdam University Medical Centers, University of Amsterdam, Cancer Center Amsterdam, Meibergdreef 9, 1105AZ Amsterdam, The Netherlands; 2Department of Molecular Genetics, Oncode Institute, Erasmus MC Cancer Institute, Erasmus University Medical Center, Doctor Molewaterplein 40, 3015GD Rotterdam, The Netherlands; y.liang@erasmusmc.nl (Y.L.); r.kanaar@erasmusmc.nl (R.K.); 3Department of Radiation Oncology, Amsterdam University Medical Centers, University of Amsterdam, Cancer Center Amsterdam, Meibergdreef 9, 1105AZ Amsterdam, The Netherlands; h.crezee@amsterdamumc.nl

**Keywords:** hyperthermia, heat stress, heat shock response

## Abstract

**Simple Summary:**

Hyperthermia is a method to expose a tumor to elevated temperatures. Heating of the tumor promotes the effects of various treatment regimens that are based on chemo and radiotherapy. Several aspects, however, limit the efficacy of hyperthermia-based treatments. This review provides an overview of the effects and limitations of hyperthermia and discusses how current drawbacks of the therapy can potentially be counteracted by inhibiting the heat stress response—a mechanism that cells activate to defend themselves against hyperthermia.

**Abstract:**

Cancer treatments based on mild hyperthermia (39–43 °C, HT) are applied to a widening range of cancer types, but several factors limit their efficacy and slow down more widespread adoption. These factors include difficulties in adequate heat delivery, a short therapeutic window and the acquisition of thermotolerance by cancer cells. Here, we explore the biological effects of HT, the cellular responses to these effects and their clinically-relevant consequences. We then identify the heat stress response—the cellular defense mechanism that detects and counteracts the effects of heat—as one of the major forces limiting the efficacy of HT-based therapies and propose targeting this mechanism as a potentially universal strategy for improving their efficacy.

## 1. Introduction

Hyperthermia (HT)—the exposure of malignant tissues to supraphysiological temperatures—is gaining popularity in clinical cancer treatment, and a wide range of HT-based strategies have been developed to achieve various clinical goals. In this review, we focus in particular on temperatures in the range of 39–43 °C, referred to as mild HT [1], which is generally applied to enhance the cytotoxic effects of chemo and radiotherapy [2,3,4]. Moreover, we focus on the sublethal effects of heat, as opposed to the effects caused by ablative hyperthermia temperatures (>45/46 °C).

Despite the clinical successes that have been achieved thus far, the efficacy of these treatments is limited by several factors including the emergence of thermotolerance—a phenotype of temporarily increased resistance to ongoing and subsequent heat exposure—in treated cells. Overcoming these limitations carries the promise of improving the clinical outcomes, irrespective of tumor or treatment type. In this review, we provide an overview of the biological effects of HT and their clinical implications. We then describe the various strategies by which cells detect heat stress and activate the transcriptional program known as the heat stress response. Finally, we discuss how the activation of this mechanism may be driving some of the important limitations of clinical HT-based treatments and conclude that targeting this mechanism may potentially be a universal strategy to improve their efficacy.

## 2. Applications and Limitations of Hyperthermia-Based Cancer Treatments

### 2.1. The Origins of Hyperthermia in Cancer Treatment

HT is among the oldest existing methods in cancer treatment. Records dating back to around 3000 B.C. describe the use of heat to burn tumor masses by a process known as cauterization in ancient India, Egypt, and China [5], along with the use of natural heat sources (e.g., hot mud baths and volcanic steam) for general wellbeing. Moreover, illustrious philosophers of ancient Greece and Rome (500 B.C.–50 A.D.), such as Parmenides, Hippocrates and Celsus, shared the opinion that manipulating body temperature is the key to curing cancer, at least when other treatment options fail [6]. The origins of the modern HT treatments, however, date back to the 1850s when surgeons William B. Coley and Carl D.W. Busch observed that in some cases the severity of fever positively correlates with tumor regression and overall patient survival. Moreover, artificial induction of fever by the injection of a mixture of dead bacterial strains, nowadays known as “Coley’s toxin”, could lead to similar outcomes [7]. Indeed, as the immune response against Coley’s toxin likely contributed to the observed responses, Coley’s findings not only marked the beginning of modern HT therapy, but are also considered pioneering work in the field of immunotherapy. Although technical difficulties related to heating and temperature monitoring [4] temporarily prevented HT from competing with the gold standards in cancer treatment (i.e., surgery, chemotherapy, and radiotherapy), different forms of HT therapies are now gaining popularity as standalone treatments, or in combination with other therapeutic agents.

### 2.2. Clinical Application of Hyperthermia

Three parameters are particularly relevant in HT treatment protocols. First is the amount of applied heat or, more specifically, the thermal dose, which depends on both the achieved temperature and the duration of heating. Second is the size of the heated body region, with whole-body, (loco-)regional and local HT referring to the application of heat to the entire patient body, a specific region (e.g., a body cavity, limb, or organ) or only the tumor, respectively [1,8] (Table 1). Third, relatively understudied and underappreciated, is the kinetics of heat application: results of some interesting studies suggest that at a constant total thermal dose, a steep temperature increase can be considerably more cytotoxic than gradual heating [9,10].

The physiological and cellular effects caused by HT are largely temperature dependent [11], providing opportunities for achieving distinct clinical goals. Temperatures exceeding 50 °C are generally used to cause direct thermal ablation in small solid neoplasms, e.g., in the kidneys, liver, lungs, prostate and bones [8]. Mild HT, on the other hand, is applied locally or regionally to promote the effects of chemo and radiotherapy on a variety of tumors, including breast, cervix, head & neck, soft tissue sarcoma, rectum and bladder [2,3,4]. The external methods to induce mild HT for superficial and deep-seated tumors include radiative techniques employing electromagnetic (e.g., radiofrequency, microwaves or infrared) or acoustic waves (e.g., ultrasound) [1,8]. The internal techniques include intraluminal and interstitial mild HT and involve placing heating probes in or near the tumor [1,8]. Regional mild HT can also be established by isolated perfusion of a part of the body, whereby a section of the blood circulation is rerouted through an external heating device [1,12], or by circulating warm liquids containing chemotherapy in body cavities, as performed in hyperthermic intraperitoneal chemotherapy (HIPEC) and hyperthermic intravesical chemotherapy (HIVEC) for the peritoneal cavity and the bladder, respectively [12,13,14]. In addition to these routinely used clinical approaches, experimental strategies for optimizing and integrating mild HT therapies with other modalities are under investigation. For example, in magnetic fluid HT, multifunctional magnetic nanoparticles are selectively targeted to tissues, generate efficient local tumor heating when exposed to an alternating magnetic field and carry different appropriate payloads [15,16,17]. In addition, the development of thermosensitive liposomal drug carriers allows for controlled drug release in tumor regions subjected to mild HT [18,19,20]. Thus, a wide range of methods have been developed to achieve heat delivery, and ongoing efforts could potentially improve the clinical applicability and versatility of mild HT.

### 2.3. Limitations of Hyperthermia-Based Therapies

Clinical successes have been achieved by utilizing mild HT as a chemo or radiosensitizing agent [2,3,4], but the efficacy of these treatments is nevertheless limited by a number of factors, irrespective of the treated tumor subtype. Firstly, to efficiently sensitize a tumor to chemo or radiotherapy, an adequate thermal dose must be delivered to the entire tumor volume while minimizing the temperature increase in the surrounding noncancerous tissues [21]. Higher tumor temperatures are associated with better clinical outcomes [22,23]. Due to both technical and biological reasons, however, achieving these goals remains challenging. For example, well vascularized subvolumes within the tumor can result in local “cool spots”, and excessive temperatures (“hot spots”) can occur in poorly vascularized normal tissue regions outside the tumor or at the interface between different tissue types. These hot spots must be suppressed as they cause discomfort and, potentially, toxicity to the patient, and clinicians are often forced to decrease the power delivered to the tumor to the extent that the desired target temperature or thermal dose cannot be achieved [24]. Secondly, most sensitizing effects of mild HT are of a temporary nature, resulting in a relatively short time window during which the tumor is optimally sensitized to the coadministered treatment modalities. This is illustrated, for example, by the finding that a long time-interval between HT and radiotherapy negatively affects the overall survival of women with locally advanced cervical cancer [25], but a debate is ongoing to what extent these findings are patient and treatment dependent [26,27,28]. Thirdly, heat stress triggers a protective cellular response, known as the heat stress response, which counteracts the effects of HT and temporarily renders cells less sensitive or even insensitive (i.e., thermotolerant) to a subsequent treatment, thereby reducing its chemo and radiosensitizing effects [29,30]. Consequently, there is a practical limit on the frequency at which HT can be applied, such that HT sessions are scheduled at intervals of at least three days. Finally, the ability of HT-based therapies to sensitize metastases, in addition to the primary tumor, is currently restricted to the local metastases that are targeted in regional HT. In conclusion, there are important limitations to the efficacy of clinical mild HT-based therapies and overcoming these limitations could improve their clinical performance and patient outcomes.

**Table 1 cancers-13-01243-t001:** Clinical application of mild hyperthermia (HT). An overview of the most common indications and the equipment used.

Region of Exposure	Tumor Stage or Position	Mode of Clinical Application	Cancer Types Treated	Clinical Studies, Technical Literature [1,8]
Local	Superficial	Acoustic waves (e.g., Ultrasound) Electromagnetic waves (radiofrequency, microwave, infrared)	Breast cancer Soft tissue sarcoma Head & neck cancer Malignant melanoma	Vernon et al. (1996) [31]
				Issels et al. (2018) [32]
	Deep seated	Intraluminal Interstitial Acoustic waves		Valdagni et al. (1994) [33]
				Overgaard et al. (1995) [34]
Regional	Deep seated	Electromagnetic waves Hyperthermic intravesical chemotherapy (HIVEC)	Cervix cancer Bladder cancer Ovarian cancer Prostate cancer Pancreatic cancer Rectal cancer Soft Tissue Sarcoma Malignant melanoma Pseudomyxoma peritonei Peritoneal mesothelioma Primary peritoneal carcinoma Gastric cancer Colon cancer	van der Zee et al. (2000) [35]
	Local metastasis	Perfusion Hyperthermic intraperitoneal chemotherapy (HIPEC)		Colombo et al. (2011) [36]
				Maluta et al. (2007) [37]
				van der Horst et al. (2018) [38]
				Wust et al. (2002) [39]
				Issels et al. (2018) [32]
				Eggermont et al. (2003) [40]
				Koops et al. (1998) [41]
				Kusamura et al. (2021) [42]
				Goéré et al. (2017) [43]
				van Driel et al. (2018) [12]
				Verwaal et al. (2003) [44]
Whole-body	Distant metastasis	Thermal chambers	Malignant melanoma	Lassche et al. 2019 [45]

## 3. The Biological Effects of Hyperthermia

### 3.1. The Effects of Hyperthermia at the Macroscopic Level

#### 3.1.1. Promotion of Tumor Perfusion and Alteration of the Tumor Microenvironment

The tumor microenvironment (TME) encompasses all components of a tumor that are not cancer cells, such as the vasculature, the extracellular matrix, various cell types (e.g., immune cells, cancer-associated fibroblasts and tumor-associated endothelial cells), as well as the factors that they secrete (e.g., cytokines, growth factors) [46,47]. It has become clear that the TME plays a considerable role in many aspects of cancer progression [46] and aids in the development of therapy resistance [47]. An important feature of the TME is the presence of hypoxic, acidic and nutrient-deprived regions, as their existence and extent positively correlate with drug resistance, metastatic potential and poor prognosis [47,48,49,50]. These regions generally arise due to difficulties in establishing a homogenous blood supply throughout the tumor volume.

Mild HT has the ability, however, to alter many of these microenvironmental parameters, as temperatures of 39–43 °C maintained for 30–60 min have generally been found to increase perfusion for the subsequent 4–8 h [51] in both well and poorly-vascularized regions of solid tumors [52,53,54,55]. This leads to normalization of oxygen, nutrient and pH levels. Although this effect was already observed in the 1980s [56,57,58], enhanced tumor perfusion and oxygenation are still thought to be the predominant mechanisms by which mild HT improves chemo and radiotherapeutic treatments [4,59,60]. Accordingly, mild HT boosts the sensitivity of tumors to agents such as radio and chemotherapy, which require oxygen for maximum cytotoxicity [61,62,63]. In addition, the increase in tumor perfusion can locally increase drug concentrations at the tumor site. Although this is still under debate for nonencapsulated drugs, various studies have convincingly shown that the extravasation of liposomal drug carriers increases upon mild HT, in part by increasing the size of blood vessel pores, which allows their escape from the circulation [18,51,64,65]. Increased tumor perfusion and the subsequent alterations of the TME are thus underlying multiple beneficial effects of mild HT-based treatments.

#### 3.1.2. Immunostimulatory Effects

Antitumor immunity plays a key role in cancer, and intensive investigation during the last decades have led to insights into the contribution of a wide range of immune cells to this effect [66,67]. The interaction between the tumor and the immune system is a constantly evolving process, wherein the immune system attempts to eliminate transformed cells as the tumor executes various countermeasures, such as attracting immunosuppressive cells and inhibiting the function of others [68]. In line with the notion that the thermal component of fever plays an important role in immune activation [69], an increasing body of data suggests that both mild HT and thermal ablation therapies can tip the delicate balance between the tumor and the immune system, in favor of the latter [70,71].

Several effects provoked by mild HT contribute to this phenomenon. Heat has been shown, for instance, to enhance immune cell recruitment by stimulating perfusion and circulation in the tumor, along with the upregulation of adhesion molecules of the vasculature, such as ICAM-1, which promote immune cell recruitment [72]. The activity of various immune cells is also increased by heat directly [69], and by the altered activity of proteins on the surface of immune and tumor cells [71]. Furthermore, the release of immunostimulatory factors by cancer cells, including exosomes [73,74] and tumor antigen-bound chaperone proteins [75,76], is enhanced by heat stress. In line with these findings, multiple preclinical studies have shown that the reduction in tumor growth and prolonged survival after mild HT treatments positively correlate with the enhanced infiltration and activation of immunity-promoting cells (e.g., APC, NK, CD4^+^ T and CD8^+^ T cells), and with higher concentrations of proinflammatory cytokines and other immune-stimulating factors [77,78,79,80]. In short, important local and systemic effects of mild HT are exerted by boosting antitumor immunity, which might be further exploited in novel combination therapies that aim to promote antitumor immune responses, including immune checkpoint inhibition [81] and chimeric antigen receptor (CAR) T-cell therapy [82].

### 3.2. The Effects of Hyperthermia at the Cellular Level

#### 3.2.1. Effects on the Folding and Structure of Proteins and Lipids

Exposure to heat has wide-ranging effects on the cell (Figure 1), mainly by affecting the structure of proteins and lipids. Most proteins have evolved to operate at the optimal growth temperature of the host organism [83]. A deviation from this temperature can cause protein unfolding and aggregation in virtually all cellular compartments [84,85], thereby affecting the functioning of the corresponding cellular pathways. Aggregates can, however, also immobilize fully functional proteins by a principle known as coaggregation. For this reason, the effects of heat-induced protein unfolding are not limited to the cellular processes that these proteins are involved in. In addition, the alteration of lipid structures translates into enhanced permeability of various cellular membranes. Ultimately, heat can result in a cell cycle halt in any phase [86], followed by cell death or senescence if the amount of damage exceeds the capacity of cellular countermeasures. In the remainder of this section, we focus on the cellular effects of mild HT that are currently believed to be most consequential for cell survival.

#### 3.2.2. Heat Alters Membrane Characteristics and Promotes Drug Influx

Elevated temperature increases the fluidity, and therefore the permeability, of cellular membranes. This can lead to a loss of ion (e.g., Ca^2+^, K^+^, Mg^2+^) homeostasis and a decline in cytosolic pH [87,88,89,90]. These alterations, however, are not necessarily all detrimental, as has been suggested for Ca^2+^ [89]. In contrast, a well-known consequence is the loss of the proton gradient over the mitochondrial inner membrane (i.e., mitochondrial uncoupling), leading to a temporary drop in mitochondrial ATP production. In addition, mitochondrial uncoupling leads to enhanced production of reactive oxygen species (ROS) by these organelles [91,92,93,94], including under hyperthermic conditions [95,96,97,98]. By transferring electrons to various cellular components, including DNA, proteins and lipids, ROS can then cause widespread detrimental effects on various cellular components, and especially on genetic material [99].

The permeability of membranes towards various compounds is also enhanced, and several in vitro studies suggest that this could contribute to the effects of certain chemothermotherapy regimens [100,101,102]. In some cases, mild HT can promote the active uptake of drugs. Heat was shown, for example, to activate CTR1, the copper transporter that is known to play a role in the uptake of cisplatin [103]. Mild HT can thus alter the characteristics of membranes, thereby interfering with various cellular processes and sensitizing cancer cells to other treatment modalities.

#### 3.2.3. Cytoskeletal Defects Further Impede the Function of Organelles

Another major consequence of heat exposure is the collapse of the cytoskeleton, governed by the unfolding and aggregation of filament forming proteins (e.g., actin, tubulin, and intermediate filaments) [104]. This leads to the deregulation of cellular processes that heavily rely on the cytoskeleton (e.g., vesicular transport, chromosome segregation, cell-cell communication, migration). Combined with the alterations in membrane properties and protein integrity, cytoskeletal dysfunction results in the disruption of the localization and, therefore, the functioning of organelles [86,105,106]. Heat stress causes, for instance, a drop in translation and protein synthesis due to fragmentation and impairment of the ER-Golgi apparatus. Finally, impaired cytoskeleton function reduces the number of mitochondria, further disturbing energy production. In short, the collapse of the cytoskeleton by mild HT contributes to the dysregulation of multiple other cellular functions.

#### 3.2.4. Hyperthermia Interferes with Nuclear Processes and Sensitizes Cells to DNA-Damaging Agents

Of all organelles, the effects of heat on the nucleus have been studied most intensively. It is a highly vulnerable structure, as even temperatures as low as 40 °C cause denaturation of some nuclear proteins [107]. This is partially due to the relatively high protein and DNA concentration, which can lead to rapid coaggregation once protein denaturation occurs [96]. Heat stress has additionally been shown to cause a massive redistribution of proteins to the nuclear matrix, and to alter the import and export of proteins [108,109,110,111], thus interfering with a wide range of nuclear processes. For instance, exposure to mild HT leads to a short-lived halt in DNA replication by impeding replicon initiation, chromatin maturation and replication fork activity [86,112]. In addition, heat interferes with transcription and RNA processing [113,114]. These effects are, however, not irreversible, allowing for the activation of transcriptional programs that are, as will be discussed in later sections, part of the heat stress response. The ability of heat to interfere with the function of various DNA repair mechanisms (e.g., homology-directed repair, nonhomologous end-joining, base-excision repair) are known to greatly contribute to the efficacy of DNA-damaging therapies, including chemo and radiotherapy [115]. Heat may further amplify these effects by inducing various types of DNA damage, including nucleotide modifications and even, albeit controversially, DNA double-strand breaks (DSBs) [112,116,117,118]. In conclusion, mild HT affects various nuclear processes and sensitizes tumor cells to DNA-damaging therapies.

## 4. Response and Adaptation to Hyperthermia

### 4.1. Detection of Hyperthermia

#### 4.1.1. The Heat Stress Response Is Activated in Response to Hyperthermia

Cells predominantly sense heat and other forms of stress (e.g., oxidative, pH, hypoxic, toxin, replication, mutation and metabolic stress) in an indirect manner by monitoring the integrity of proteins. Most cellular compartments contain a dedicated mechanism that communicates heat stress to the nucleus, leading to the activation of a transcriptional program that counteracts its effects, known as the heat stress response.

In the cytosol, most forms of stress cause unfolding of proteins, resulting in the activation of the “heat shock response” [119]. A hallmark of the heat shock response is the induction of heat shock proteins (HSPs) [120,121]. HSPs are a subgroup of molecular chaperones that are involved in a myriad of functions by folding and refolding various “client” proteins [122]. HSPs protect animals from oxidative stress, endotoxin-mediated microvascular injury and ischemic heart damage, among various other important non heat-related functions [122]. As heat stress triggers protein unfolding, HSPs play an important role in the heat shock response.

Although novel mediators are continuously being uncovered, the family of heat shock factors (HSFs) is known to play an important role in the onset of the heat shock response [123]. HSF1 seems to be a key player, as its deletion in mice completely abolishes the induction of HSPs [124,125]. In the absence of heat stress, HSF1 is kept inactive by intramolecular interactions and association with various (co)chaperones (e.g., HSP90, HSP70, and HSP40). Heat stress, however, leads to the increased affinity of these chaperones towards other (unfolded) proteins, thereby releasing HSF1. Under the regulation of numerous post-transcriptional modifications [126,127], HSF1 forms homotrimers, translocates to the nucleus and stimulates transcription by binding to the consensus heat shock elements (HSEs). After the cell recovers from heat stress, similar but inverse processes are involved in returning HSF1 to its inactive state. Although HSF1 has long been regarded as an essential transcription factor in the overall activation of the heat shock response, recent studies revealed that its role may be limited to the induction of HSPs, whereas other transcription factors, including serum response factor (SRF) [128], orchestrate the bulk of the response [129,130]. In addition, accumulating evidence indicates a modifying role of other HSFs, such as HSF2, HSF3, and HSF4 [124,125]. For instance, although HSF2 depends on HSF1 for its role in the heat shock response, it has been shown that HSF2 can also bind to HSEs and interacts with HSF1 [131]. Currently available evidence seems to neither confirm nor deny the involvement of additional HSFs, such as HSF5, HSFX, and HSFY [127]. It is also noteworthy that major modulators of the heat shock response, such as the Hippo signaling pathway, have eluded discovery until quite recently [132]. Another signaling cascade that is activated in response to (heat) stress is the pathway of the stress response kinases SAPK and p38 [133]. Triggered at the cellular membrane by receptors such as TNFα, IL1β or G-protein coupled receptors, the pathway involves a complex, multilevel, and multiprotein cascade that finally activates SAPK and p38 and, in turn, Elk1, cJun and ATF-2 transcription factors. Downstream effector processes include adaptation to stress, but also immune system activation, inflammation and apoptotic responses. Interestingly, HSP70 can prevent activation of this cascade, and especially its apoptosis-stimulating effects [134,135], thereby contributing to stress-resistance.

Similar to the cytosol, compromised integrity of proteins in organelles results in the activation of the so-called “unfolded protein response” (UPR), and is well-characterized for the ER-Golgi system (UPR^ER^) [136] and mitochondria (UPR^mt^) [137,138]. The ER is the primary site for synthesis and modification of proteins, and protein integrity is, therefore, vital for its function. HT and other forms of stress, however, lead to protein unfolding in the ER [139]. Interestingly, short pre-exposure to heat has been shown to reduce ER stress upon subsequent exposure [140], suggesting that recovery of ER homeostasis is an important feature of the UPR. ER stress is sensed by transmembrane proteins IRE1, PERK, and ATF6. Although these proteins activate separate signaling pathways, crosstalk between the pathways is evident, and their activation eventually leads to the triggering and nuclear translocation of ATF6, XBP1 and eIF2α. Subsequently, the repertoire of UPR^ER^ induced genes relieves ER stress by enhanced chaperone activity, removal of damaged ER by ER-associated protein degradation (ERAD) and expansion of the ER by promoting lipid synthesis [136]. In mitochondria, the UPR^mt^ is activated when the level of misfolded proteins exceeds the capacity of chaperones in the mitochondrial matrix. In yeast, this excess of proteins is degraded by the protease ClpXP, allowing for their transport to the cytosol via the peptide transporter HAF-1. Subsequently, the presence of peptides triggers activation and nuclear translocation of transcription factor ZC376.7. Studies in yeast are, however, difficult to translate into the context of higher organisms, and which proteins communicate mitochondrial protein stress to the cytosol in mammalian cells remains elusive. It is known, however, that JNK2 activation and translocation to the nucleus is an important consequence, as JNK2 phosphorylates the Jun transcription factor, which promotes expression of CHOP and C/EBPβ. These transcription factors, together with others that are yet to be defined, drive expression of mitochondrial chaperones [137,141]. To conclude, the monitoring of protein integrity plays a key role in the detection of heat and in the activation of pathways that are collectively forming the heat stress response (Figure 2).

**Figure 2 cancers-13-01243-f002:**
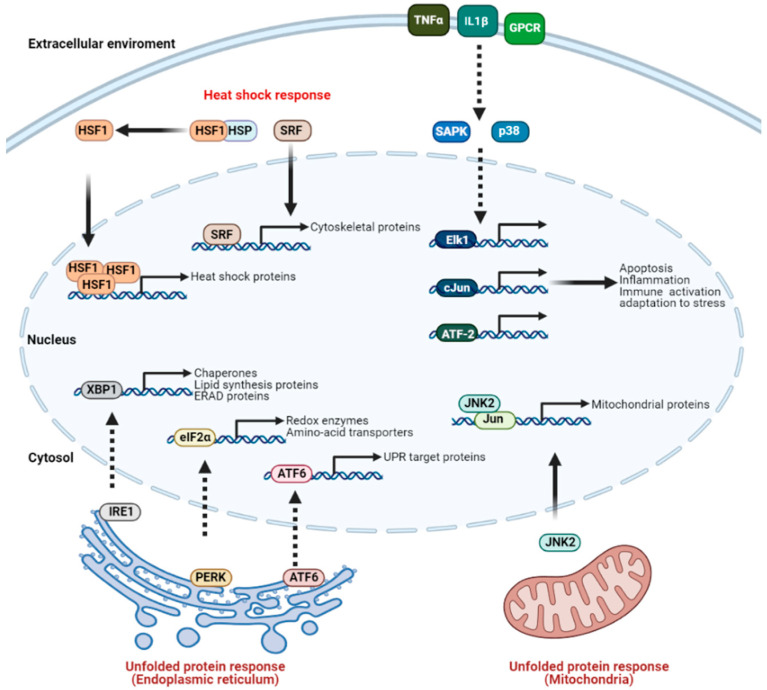
Activation and functions of the heat stress response. Heat stress is mainly sensed by monitoring protein integrity. The unfolded protein response (UPR) is activated when heat stress is detected in organelles [136,137], whereas the heat shock response communicates cytosolic heat stress [119,126]. These pathways, collectively termed the heat stress response, initiate cellular countermeasures that drive heat adaptation. Solid and dashed arrows indicate direct and indirect involvement, respectively. Figure adapted and extended from [142].

#### 4.1.2. Thermosensors as Direct Regulatory Mediators of Thermal Stress

Although it was assumed at first that activation of the heat stress response is initiated solely by the detection of unfolded proteins, it has become apparent that it can also be activated in their absence. Cells contain a set of biomolecules, often referred to as thermosensors, which can be nucleic acid-, protein- or membrane-based. The conformational changes of these molecules in response to heat are directly coupled to the regulation of elements of the heat stress response. Most studies initially focused on the role of prokaryotic thermosensors in the regulation of virulence genes to obtain insights in their pathogenicity [143,144,145]. Recent studies, however, provide evidence for the existence of similar mechanisms in eukaryotes. For instance, alterations in membrane fluidity, comparable to the alterations at febrile temperatures, have been shown to activate the heat stress response in the absence of unfolded proteins [146,147]. Although we have just started to unravel the functioning of these mechanisms, they might provide novel therapeutic opportunities in combination therapies that involve heat application.

#### 4.1.3. Activation of the Heat Stress Response on the Systemic Level

Next to the cell-autonomous mechanisms described above, a growing body of evidence suggests that eukaryotes communicate and react to heat stress on a systemic level. It has been shown in *Caenorhabditis elegans,* for instance, that heated tissues excrete various stress factors and trigger neuronal signaling, thereby initiating the activation of the heat stress response in distant body regions [148,149,150]. With some emerging studies demonstrating the interplay between tumors and the nervous system [151], more research is needed to elucidate whether and how the tumor heat stress response is affected by systemic responses.

### 4.2. The Heat Stress Response Activates Various Counteractive Measures

Studies focusing on the heat stress response in various species have led to the consensus that exposure to elevated temperatures results in the induction of seven functionally distinct classes of proteins originating from 50–200 genes [152,153,154,155,156,157,158,159]. Conversely, heat stress also results in the temporary repression of thousands of genes, mostly involved in transcription, translation, cell growth and several aspects of RNA processing [129,142,160,161]. Furthermore, accumulating evidence suggests an important role for post-translational modifications in the regulation of the heat stress response [162]. Molecular chaperones, such as HSPs, belong to the most upregulated class, and their expression is initiated within minutes. Following suit are proteins involved in the proteolytic system, including proteases and components of the proteasome. Together, molecular chaperones and the proteolytic system ensure that misfolded and (co)aggregated proteins either undergo refolding or are degraded, thereby facilitating rapid recovery of proteostasis. Other upregulated protein classes respond to more specific forms of damage. For example, DNA/RNA repair/modifying enzymes are upregulated to counteract the induction of DNA damage, replication stress and the initial drop in transcription and translation. Components and regulators of the cytoskeleton are overexpressed to counteract the detrimental effects of heat on the cytoskeleton, and subsets of metabolic enzymes are overexpressed to accommodate for the reduced energy production. In short, the heat stress response regulates the expression of a wide range of processes to counterbalance the detrimental effects of heat.

### 4.3. Thermotolerance

The phenotype emerging from all the discussed responses to heat stress—thermotolerance—provides a degree of limited and temporary adaptation to the ongoing and/or subsequent exposures. Thermotolerance can be manifested in multiple complex ways on the level of cells, tissues and organs, but a common denominator is an increased resistance to the stressor, which is generally proportional to its magnitude [163]. In the context of clinical HT, the phenomenon complicates and restricts treatment scheduling and reduces efficacy [164]. The kinetics of acquisition and persistence of the thermotolerant state strongly depend on the activation of the heat stress response pathways and, despite their complexity discussed above, thermotolerance can be modelled based on network analysis of HSP upregulation alone, at least in in vitro experiments [9].

## 5. Optimizing Hyperthermia-Based Treatments by Manipulating the Heat Stress Response

### 5.1. Inhibition of the Heat Stress Response Enhances the Efficacy of Hyperthermia-Based Treatments in Preclinical Research

As discussed in the earlier sections, therapies based on mild HT frequently fail to reach the required thermal dose and offer a short therapeutic window, while the acquisition of thermotolerance limits the frequency at which they can be applied. Since the ability to generate an adequate heat stress response is a prerequisite for (cancer) cells to survive exposure to heat, it can be hypothesized that interfering with this process will improve treatment efficacy. Accordingly, a lower thermal dose may elicit similar cytotoxic effects, the duration of the therapeutic window may be extended and thermotolerance may be reduced or eliminated, offering opportunities to increase the treatment frequency. Various studies indeed support the notion that the acquisition of thermotolerance can be counteracted. For instance, mouse embryonic fibroblasts lacking HSF1 have been shown to be incapable of generating the thermotolerant phenotype, whereas heat preconditioning of the wildtype counterpart did result in a reduced sensitivity to subsequent HT [165]. Disruption of HSF1 function by lentiviral overexpression of its dominant-negative counterpart has, similarly, been shown to sensitize the highly thermotolerant Bcap37 breast cancer cell line to mild HT [166]. Next to these genetic approaches, others have demonstrated that the pharmacological inhibition of various HSPs by different compounds, such as STA-9090 (also known as ganetespib, targeting HSP90), quercetin, and KNK437 (targeting HSP70 and HSF1), impedes the acquisition of a thermotolerant state by cervix, prostate and colon cancer cells [167,168,169]. Furthermore, it has been shown in a squamous cell carcinoma mouse model that heat-preconditioned tumors can be resensitized to mild HT by pharmacologically inhibiting the activation and the function of HSPs by KNK437 [170]. The notion that the suppression of the heat stress response could allow for lower thermal dosages to be effective is also supported by recent studies that achieved a similar reduction in cell survival and tumor growth with milder heat treatment schedules when the function of chaperones such as HSP90 and HSP70 was pharmacologically inhibited [167,171]. In line with the potential of heat stress response intervention to reduce the drawbacks of mild HT-based therapies, all studies mentioned above, as well as others, reported an increased efficacy of heat treatments that were combined with manipulation of the heat stress response as compared to heat alone. Pharmacological inhibition by quercetin, Pifithrin-μ, or RNAi-mediated knockdown of HSP70 and HSF1, led to enhanced cell killing of prostate and melanoma cancer cells in vitro and in vivo [172,173,174]. Further, the combined inhibition of HSP90, HSP70, and HSF1 by 17-DMAG and quercetin enhanced the treatment efficacy of magnetic nanoparticle-mediated HT treatment in a melanoma mouse model [175]. Implementation of HSP90 inhibition in a chemo and radiotherapy-based mild HT treatment regimen did result in enhanced treatment efficacy in cervix cancer cells [167]. Thus, accumulating evidence indicates that the heat stress response may be responsible for many major drawbacks of mild HT-based treatments, and that inhibition of this response could reduce these drawbacks, as well as improve the overall performance (Table 2).

### 5.2. Inhibiting the Heat Stress Response in Hyperthermia-Based Therapies: Opportunities and Challenges

The past decades yielded valuable insights into the tumor-supporting effects of stress response mechanisms, and especially of the molecular chaperones. These have been shown to drive tumor progression by chaperoning oncoproteins, thereby promoting cancer progression in multiple aspects, including elevated stress tolerance, enhanced proliferation, escape from cell death, induction of the epithelial-to-mesenchymal transition, migration and invasion [122,176]. Driven by these insights, a considerable effort has been invested into the development of specific inhibitors of HSPs, such as HSP90 and HSP70, and promising compounds displaying favorable toxicity profiles are currently being evaluated for their use against a multitude of different cancer types [177,178]. Additionally, potent inhibitors of transcriptional regulators of the heat stress response, such as HSF1, have been generated [179], but their current generations lack potency and specificity and display unfavorable toxicity profiles in vivo [180,181]. Although these targeted agents are applied systemically, potentially exposing patients to unwanted side-effects, the chaperone addiction shared by most advanced tumors makes them considerably more sensitive than normal tissues, yielding a certain degree of specificity [182,183,184]. But, perhaps more importantly, in the context of HT the specificity is a result of the local nature of most currently applied HT-based treatments. Unfortunately, however, the mixed outcomes of clinical trials testing heat stress response inhibitors have, thus far, precluded their regulatory approval, reducing the chance for fast clinical adaptation.

A deeper understanding of the heat stress response is required to develop new selective inhibitors of the heat stress response because a large portion of our knowledge is derived from studying plants, bacteria and fungi. Although such studies have led to valuable insights into the core functions of the heat stress response, e.g., prevention of protein dysfunction, these findings cannot be directly translated to the clinical HT context for several reasons. Firstly, as different organisms display varying sensitivity to heat [83], and as its effects show a clear dose-dependence in many cases, such as its effects on DNA repair [115], the various heating conditions used in these studies are incompatible with mild HT-based therapies. Secondly, the functional scope of the heat stress response can differ per organism. For instance, whereas HSF1 is only activated in response to stress in mammalian cells, it has recently been shown that yeast, a model organism that is intensively used in heat stress-related research, has a need for constitutively active HSF1 to prevent protein aggregation [185], yielding it incomparable to many higher organism models. Thirdly, a growing body of evidence suggests that cellular stress responses, including the heat stress response, are differently organized and take on noncanonical functions in several processes in the context of cancer. For example, HSF1 has been shown to play a more pronounced role in pathways responsible for energy metabolism, DNA repair, cell cycle signaling and apoptosis in colon, breast and lung tumors [154]. Importantly, the degree by which the heat stress response is activated, and the extent by which it takes on these other roles, are correlated with tumor stage, metastasis and patient survival [154,176]. These findings reveal an extra layer of complexity, but also provide a basis for understanding why potent HSP inhibitors have only produced promising clinical results under very specific conditions, where tumors showed a dependency on these chaperones for the maintenance of particular oncoproteins. For instance, recent studies showed that chaperones in a subset of tumors may display a higher degree of functional overlap [186]. This increases the capacity of the chaperone system, thereby enabling the tumor to thrive in a highly stressful environment, either caused by external factors or by the high mutational burden leading to misfolded proteins. This carries the drawback, however, of a diminished redundancy of the system, rendering these tumors highly sensitive to inhibition of individual chaperones and their cofactors [186]. In retrospect, these findings offered a convincing explanation for the disappointing clinical results that have been obtained with, for instance, HSP90 inhibitors [187]. They also underscored the importance of understanding how heat stress response is regulated in the context of cancer. In summary, the heat stress response appears to drive some of the important mechanisms that limit the efficacy of therapies based on mild HT, but more research is clearly needed to efficiently exploit these mechanisms in clinical settings.

## 6. Conclusions and Future Perspectives

Treatments based on mild HT are applied to a widening range of cancer types. In the clinic, however, their efficacy is often limited by the insufficient thermal dose delivery, the short therapeutic window and the acquisition of thermotolerance, which limits treatment frequency. The evidence presented in this review suggests that a common mechanism—the cellular heat stress response—may be underlying many of these limitations (Figure 3), and that mild HT combination therapies that target this mechanism might improve the clinical outcomes. The good news is that some heat shock response inhibitors, especially those targeting HSP90 (e.g., ganetespib), have shown promise in dozens of preclinical studies and displayed favorable pharmacological profiles in advanced, large, phase II and III clinical trials. Considerable challenges remain, however. First, none of these drugs have been approved for use by regulatory authorities, and this is unlikely to change in the short term. Second, the safety and efficacy of HT combination therapies needs to be demonstrated in new trials, which are generally slow in patient accrual. This could be offset by multicenter studies, but the local differences in HT application protocols make such an approach difficult as well. Third, due to the currently overall low number of patients eligible for HT treatments, and thus the limited potential market for drug sale, there is relatively little interest from pharmaceutical companies in sponsoring the required costly studies.

So, what can be done to overcome these challenges? It is unlikely that late-stage clinical trials will be initiated by the HT field, but the considerable clinical interest in heat stress response inhibitors for other indications may facilitate the development of HT-based combination therapies via off-label use. Clinical translation may also be easier if the inhibitors are applied locally, rather than systemically. This can be especially attractive in HIVEC and HIPEC therapies, wherein heated chemotherapeutic cocktails are circulated in the abdominal cavity or the bladder, respectively. Locally-applied “thermosensitizers” would have a smaller chance of inducing systemic toxicity, likely accelerating the approval process. Finally, one path that is yet to be explored systematically in hyperthermic oncology is drug discovery and repurposing. To our knowledge, no large-scale compound screens have been performed to uncover new thermosensitizers, or to reveal thermosensitizing properties of existing drugs. Such screens could not only provide valuable new information on the mechanisms driving the cellular heat stress responses, but also new, safe therapeutic strategies available in the short term.

## Figures and Tables

**Figure 1 cancers-13-01243-f001:**
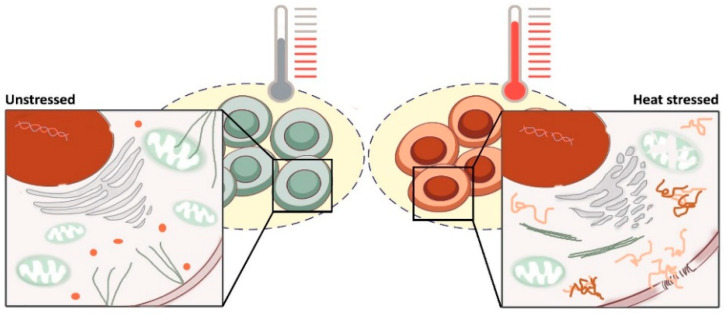
Effects of hyperthermia at the cellular level. Exposure to heat induces unfolding and aggregation of proteins (orange) and membrane permeabilization. The most detrimental consequences include cytoskeletal collapse (green) and difficulties in processes that require its function (e.g., vesicular transport, chromosome segregation, cell-cell communication, migration, maintenance of organelle structure). In addition, protein synthesis is impaired, as depicted by fragmentation of the ER-Golgi apparatus (grey). A drop in energy production arises from mitochondrial dysfunction. Furthermore, various DNA repair pathways are affected, which sensitizes tumor cells to DNA-damaging therapies.

**Figure 3 cancers-13-01243-f003:**
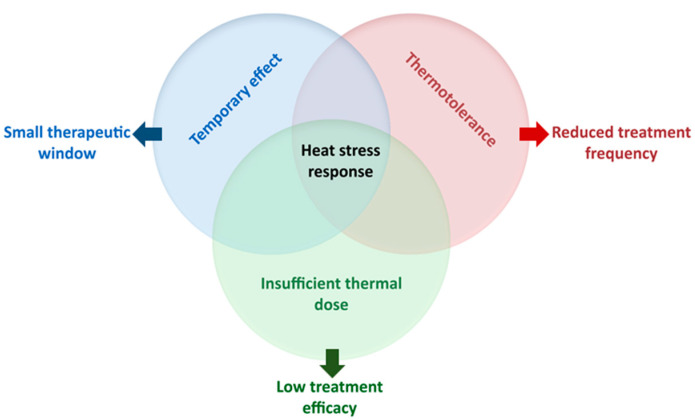
Current limitations of HT-based therapies. Venn diagram displaying the major limitations of the current mild HT-based therapies. The cellular heat stress response underlies many of these limitations, including insufficient thermal dose delivery, the short therapeutic window and the acquisition of thermotolerance.

**Table 2 cancers-13-01243-t002:** Preclinical support for combining mild HT with heat stress response inhibition. Overview of studies that successfully used heat stress response inhibition to improve the performance of mild HT-based treatments.

Study	Tissue of Interest	Target	Intervention Strategy	Biological Context	Effect on HT Treatment
Vriend et al. (2017) [167]	Cervix cancer	HSP90	Pharmacological: Ganetespib	in vitro	Enhanced treatment efficacy Disruption of thermotolerance Lower thermal dose required
Koishi et al. (2001) [170]	Squamous cell carcinoma	HSPs	Pharmacological: KNK437	in vitro	Enhanced treatment efficacy Disruption of thermotolerance
				in vivo	
McMillan et al. (1998) [165]	Untransformed	HSF1	Genetic: Knockout	in vitro	Enhanced treatment efficacy Disruption of thermotolerance
Wang et al. (2002) [166]	Breast cancer	HSF1	Genetic: Knockout	in vitro	Enhanced treatment efficacy Disruption of thermotolerance
Sahin et al. (2011) [168]	Prostate cancer	HSF1HSPs	Pharmacological: KNK437, quercetin	in vitro	Enhanced treatment efficacy Disruption of thermotolerance
Court et al. (2017) [171]	Ovarian cancer	HSP70	Pharmacological: Pifithrin-μ	in vitro	Enhanced treatment efficacy Lower thermal dose required
				in vivo	
Asea et al. (2001) [172]	Prostate cancer	HSPs	Pharmacological: Quercetin	in vitro	Enhanced treatment efficacy
				in vivo	
Sekihara et al. (2013) [173]	Prostate cancer	HSP70	Pharmacological: Pifithrin-μ	in vitro	Enhanced treatment efficacy
Yokota et al. (2000) [169]	Colon cancer	HSPs	Pharmacological: KNK437	in vitro	Enhanced treatment efficacy Disruption of thermotolerance
Nakamura et al. (2010) [174]	Melanoma	HSF1	Genetic: Knockdown	in vitro	Enhanced treatment efficacy
Miyagawa et al. (2014) [175]	Melanoma	HSPsHSF1	Pharmacological: 17-DMAG, Quercetin	in vitro	Enhanced treatment efficacy
				in vivo	

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
