# Peer review of "Modulating the Heat Stress Response to Improve Hyperthermia-Based Anticancer Treatments"

_cancers, 2021, doi:10.3390/cancers13061243_

Round 1

Reviewer 1 Report

Your paper is very interesting.

The only comment I have is concerning  table 1 when you refer to HIPEC indication yous shoud add also :

pseudomyxoma peritonei, peritoneal mesothelioma, primary peritoneal carcinoma, gastric cancer, colon cancer

Author Response

  1. The only comment I have is concerning  table 1 when you refer to HIPEC indication you should add also: pseudomyxoma peritonei, peritoneal mesothelioma, primary peritoneal carcinoma, gastric cancer, colon cancer.

We have added these indications (marked in red), as kindly suggested by the reviewer.

Reviewer 2 Report

Comments to the authors,

The review by Scutigliani et al summarized current knowledge on hyperthermia as a cancer therapy, with focus on the role of heat response pathway. While the concept of thermotolerance and thermosensetizers with respect to hyperthermia treatment has been reviewed previously, this work was interesting and had some novel summaries on the subject. Below are my specific comments which are mostly minor:

  1. The term mild hyperthermia is introduced without defining the actual temperature range.
  2. It would be helpful to include references in table 1
  3. Related to section 3.2, a recent publication by Singh et al (2020) indicated a direct effect of hyperthermia on DNA damage (should probably be referenced). While activation of the stress response is undeniable, the possibility of contributing pathways such as DNA damage response cannot be excluded and should be entertained.
  4. Section 4.1.1. does not cover the p38 pathway’s involvement in this process. A comprehensive review of literature in this regard should be carried out.
  5. In section 4.2, it would be helpful to describe thermotolerance with references. Although the molecular process leading to thermotolerance is described the term and the phenotypic resistance to high temperature with respect to the literature is not well described.
  6. The section titled: Inhibition of the heat stress response reduces major drawbacks and enhances the performance of hyperthermia-based therapies implies a review of drawbacks and optimization of the actual therapy. I would change the title to better represent the section in fact presenting mostly cell data and hyperthermia treatment as opposed to hypothermia therapy (in the clinical sense). The data only present the potential of targeting the pathways because obviously one would not be able to knock down the proteins in tumours (not in the near future at least) as done in cells.
  7. In section 5.2: Inhibiting the heat stress response in hyperthermia-based therapies: opportunities and challenges, many of the proposed drugs referenced while targeting heat response proteins, were not used in combination with hyperthermia. I think for the purpose of the review, the alternative roles of heat response pathways including in the absence of heat, and as such targeting them, or using them as hyperthermia sensitizers should be clarified and not mixed together. In this section, the specificity (or lack of) of the drug for the tumour should also be reflected upon. This would have an impact on potential use of the drug or unacceptable toxicity to normal tissue. If I am correct, only one in vivo (mouse) study was carried out using drug, heat combination? I think this section can be structured better.
  8. While in my opinion there are no grammatical issues with the language, some formulations can be modified to improve the text:

Here are just a few of examples:

- With an increasing volume of data demonstrating the interplay between tumors and the nervous system [107], future studies might elucidate the importance of the systemic organization of the heat stress response in mild HT-based treatments- somewhat ambiguous-simplify what you want to say. One reference perhaps contradictory to an increasing volume of data which has been used repeatedly.

- …a superior efficacy of heat treatments that were combined with manipulation of the heat stress response-what is meant by a superior efficacy, can this really be demonstrated using in vitro data?

- In conclusion, it is clear that the heat stress response underlies most effects that limit the efficacy of clinical therapies based on mild HT, but more re-search is needed to efficiently exploit these mechanisms- Can one really draw a conclusion from a selective review of the literature. I would for example suggest: In summary, heat stress response proteins contribute to mild HT resistance…

There are other cases of similar language concern, which should be revised in the spirit of scientific correctness. I would invariably replace terms such as major, key, large with more specific/descriptive terminology, or remove them entirely.

Author Response

  1. The term mild hyperthermia is introduced without defining the actual temperature range.

This is indeed an important and consequential term in this manuscript. In addition to the definition already present in the first paragraph of the introduction, we have now defined the term in the abstract as well.

  1. It would be helpful to include references in table 1.

We have now added relevant references to this table, which has indeed made it more useful for the readers.

  1. Related to section 3.2, a recent publication by Singh et al (2020) indicated a direct effect of hyperthermia on DNA damage (should probably be referenced). While activation of the stress response is undeniable, the possibility of contributing pathways such as DNA damage response cannot be excluded and should be entertained.

We have added the reference to this interesting paper in section 3.2.4, as suggested by the reviewer. We certainly agree that other pathways contribute to cellular protection against the effects of heat, and we have now underscored this in multiple sections throughout the manuscript.

  1. Section 4.1.1. does not cover the p38 pathway’s involvement in this process. A comprehensive review of literature in this regard should be carried out.

We appreciate this suggestion, and have added an overview of this important (but perhaps underappreciated in the context of hyperthermia) pathway to section 4.1.1 and, additionally, to updated Figure 2.

  1. In section 4.2, it would be helpful to describe thermotolerance with references. Although the molecular process leading to thermotolerance is described, the term and the phenotypic resistance to high temperature with respect to the literature is not well described.

The reviewer is correct - gathering key information about the state of thermotolerance in a single section will be useful for the reader. This is now added as a new Section 4.3.

  1. The section titled: Inhibition of the heat stress response reduces major drawbacks and enhances the performance of hyperthermia-based therapies implies a review of drawbacks and optimization of the actual therapy. I would change the title to better represent the section in fact presenting mostly cell data and hyperthermia treatment as opposed to hypothermia therapy (in the clinical sense). The data only present the potential of targeting the pathways because obviously one would not be able to knock down the proteins in tumours (not in the near future at least) as done in cells.

We agree, as most data we presented are cellular or animal studies. The title has been changed to “Inhibition of the heat stress response enhances the efficacy of hyperthermia-based treatments in preclinical research”

  1. In section 5.2: Inhibiting the heat stress response in hyperthermia-based therapies: opportunities and challenges, many of the proposed drugs referenced while targeting heat response proteins, were not used in combination with hyperthermia. I think for the purpose of the review, the alternative roles of heat response pathways including in the absence of heat, and as such targeting them, or using them as hyperthermia sensitizers should be clarified and not mixed together. In this section, the specificity (or lack of) of the drug for the tumour should also be reflected upon. This would have an impact on potential use of the drug or unacceptable toxicity to normal tissue. If I am correct, only one in vivo (mouse) study was carried out using drug, heat combination? I think this section can be structured better.

We indeed agree that information on the specificity of the inhibitors should be provided, which has now been added in section 5.2. In Table 2, column 5 (Biological context) clearly states whether data has been obtained from in vivo or in vitro experiments. Studies by Koishi et al. (2001), Court et al. (2017), Asea et al. (2001) and Miyagawa et al. (2004) were carried out in vivo.

  1. While in my opinion there are no grammatical issues with the language, some formulations can be modified to improve the text:

Here are just a few of examples:

- With an increasing volume of data demonstrating the interplay between tumors and the nervous system [107], future studies might elucidate the importance of the systemic organization of the heat stress response in mild HT-based treatments- somewhat ambiguous-simplify what you want to say. One reference perhaps contradictory to an increasing volume of data which has been used repeatedly.

- …a superior efficacy of heat treatments that were combined with manipulation of the heat stress response-what is meant by a superior efficacy, can this really be demonstrated using in vitro data?

- In conclusion, it is clear that the heat stress response underlies most effects that limit the efficacy of clinical therapies based on mild HT, but more re-search is needed to efficiently exploit these mechanisms- Can one really draw a conclusion from a selective review of the literature? I would for example suggest: In summary, heat stress response proteins contribute to mild HT resistance…

There are other cases of similar language concern, which should be revised in the spirit of scientific correctness. I would invariably replace terms such as major, key, large with more specific/descriptive terminology, or remove them entirely.

We agree with the reviewer, some of our statements were indeed somewhat grandiose, which is uncalled for. We have reviewed the entire manuscript and adjusted or eliminated these and other similar statements in numerous sections.

Reviewer 3 Report

The Ms describes the role of mild HT on anti-cancer therapies from a single sided oppinion. Many biological and especially immunological mechanism induced by heat are a bit oversimplified.

The term "mild heat" needs to be described much more clearly , effects induced by sublethal and lethal heat need to be considered comparatively. 

The term heat induced normalization of the TME needs a much more detailed explanation.

The effects of heat on protein synthesis in general and on HSP synthesis specifically need to be elaborated more detailed.

The immunological efefcts induced by heat need to be explained much more detailed. The whole chapter is too short for this important aspect.

The role of ROS (where do they come from what are they doing especially needs to be explained more detailed.

The role of heat on membrane fluidity is a transient one. This needs to be described more detailed. The statements made by the authors is an oversimplification of the biological mechanisms involved. 

It appears as if heat is the only inducer of the heat stress response. It needs to be mentioned that nearly all kinds of stress induce the heat shock response. Apart from HSF1 there are at least 8 additional HSF described which need to be mentioned.

Key literature is missing for important statements.

With respect to the HSP it is key to mention and describe more detailed that depending on the intra or extracellular localization HSPs fulfill different tasks.

The transfer of yeats data into human studies needs clarifiucation. Limitations need to be elaborated.

Table 1: Literature needs to be included.

Table 2: These are all in vitro studies . It would be much better to show data from preclinical studies. 

In general, it appears that by inhibiting the sterss response (the HSP synthesis) would solve the whole problem. This is definitely not the case.

The potentail impact of heat on novel immunotherapies such as Immune checkpoint inhibitors, CAR effector cells are completely missing.

Author Response

  1. The Ms describes the role of mild HT on anti-cancer therapies from a single sided opinion. Many biological and especially immunological mechanisms induced by heat are a bit oversimplified.

Although it is difficult to respond to this particular comment due to its general nature, in the revised version of the manuscript multiple sections have been expanded or added (especially related to the immune system) to reflect the complexity hinted at by the reviewer (see also our response to reviewers’ other comments).

  1. The term "mild heat" needs to be described much more clearly , effects induced by sublethal and lethal heat need to be considered comparatively.

This is indeed an important term in this manuscript, as also stressed by Reviewer 2. In addition to the definition that was already present in the first paragraph of the introduction, we have now defined the term in the abstract. In this manuscript we focus on sublethal effects of mild heat - we stressed this now in the introduction as well.

  1. The term heat induced normalization of the TME needs a much more detailed explanation.

We share the opinion of the reviewer that it was unclear what the term “normalization of the tumor microenvironment” means in section 3.1.1. We clarified this statement and expanded the relevant section in the revised version.

  1. The effects of heat on protein synthesis in general and on HSP synthesis specifically need to be elaborated more detailed.

Section 4.1.1 has been rewritten and expanded to emphasize the role of HSPs in the heat shock response, as suggested by the reviewer.

  1. The immunological effects induced by heat need to be explained much more detailed. The whole chapter is too short for this important aspect.

We appreciate this suggestion, we have now rewritten and expanded section 3.1.2 to provide a more detailed overview of the important roles of the immune system in HT-based therapies.

  1. The role of ROS (where do they come from and what are they doing especially needs to be explained more detailed).

We expanded section 3.2.2 to accommodate for this.

  1. The role of heat on membrane fluidity is a transient one. This needs to be described more detailed. The statements made by the authors is an oversimplification of the biological mechanisms involved.

We agree with the reviewer and now apply more nuance to section 3.2.2, both on the effects of heat on ion homeostasis as well as on permeabilization to compounds.

  1. It appears as if heat is the only inducer of the heat stress response. It needs to be mentioned that nearly all kinds of stress induce the heat shock response. Apart from HSF1 there are at least 8 additional HSF described which need to be mentioned.

We agree that it should be mentioned that nearly all kinds of stress induce the heat shock response; we modified section 4.1.1 to reflect this. We also added extra information on the involvement of other HSFs.

  1. Key literature is missing for important statements.

We have reevaluated the manuscript and added a number of important references throughout. It is likely, however, that we have not included all references that the reviewer had in mind, in which case we would very much appreciate more specific information, as we hope to make the piece as comprehensive and useful as possible.

  1. With respect to the HSP it is key to mention and describe more detailed that depending on the intra or extracellular localization HSPs fulfill different tasks.

We agree that this adds an important extra perspective on the function of HSPs and addressed this issue by stating the immunostimulatory function of extracellular HSPs in section 3.1.2, as part of our effort to provide more details on the effects of HT on the immune system.

  1. The transfer of yeast data into human studies needs clarification. Limitations need to be elaborated.

We have added or modified a number of statements throughout the manusction to underline the important limitations of the simpler model organisms, as suggested.

  1. Table 1: Literature needs to be included.

Done in the revised version.

  1. Table 2: These are all in vitro studies . It would be much better to show data from preclinical studies.

Table 2 includes all studies that we are aware of that evaluate the potential of combining hyperthermia with inhibitors of the heat stress response. Some of these studies contain preclinical in vivo data, as described in column 5. No clinical trials have been conducted, to our knowledge, that combine hyperthermia with inhibitors of the heat stress response. This is therefore the most complete presentation of the available literature. We would appreciate receiving more detailed suggestions from the reviewer in case we overlooked any data.

  1. In general, it appears that by inhibiting the stress response (the HSP synthesis) would solve the whole problem. This is definitely not the case.

We agree with the reviewer, HS inhibition is certainly not a panaceum! We have modified any statements that may have led to such conclusions throughout the manuscript.

  1. The potential impact of heat on novel immunotherapies such as Immune checkpoint inhibitors, CAR effector cells are completely missing.

This is a great suggestion, and we included relevant information in the expanded section 3.1.2.

Round 2

Reviewer 2 Report

I believe the authors have adequately met the comments and therefore recommend the manuscript for publication.

Reviewer 3 Report

In general the authors adressed all major concerns whcih were raised.